# Metric-induced nonhermitian physics

**Pasquale Marra**

Department of Engineering and Applied Sciences, Sophia University, 7-1 Kioi-cho, Chiyoda-ku, Tokyo 102-8554, Japan
Department of Physics & Research and Education Center for Natural Sciences, Keio University, 4-1-1 Hiyoshi, Yokohama, Kanagawa, 223-8521, Japan
Graduate School of Informatics, Nagoya University, Furo-cho, Chikusa-Ku, Nagoya, 464-8601, Japan

pasquale.marra@keio.jp

## Abstract

I consider the longstanding issue of the hermiticity of the Dirac equation in curved spacetime. Instead of imposing hermiticity by adding ad hoc terms, I renormalize the field by a scaling function, which is related to the determinant of the metric, and then regularize the renormalized field on a discrete lattice. I found that, for time-independent and diagonal (or conformally flat) coordinates, the Dirac equation returns a pseudohermitian (i.e., $\mathcal{PT}$-symmetric) Hamiltonian when properly regularized on the lattice. Notably, the $\mathcal{PT}$-symmetry is unbroken, assuring a real energy spectrum with unitary time evolution. Conversely, time-dependent spacetime coordinates break pseudohermiticity and return nonhermitian Hamiltonians with nonunitary time evolution. Similarly, space-dependent spacetime coordinates lead to the nonhermitian skin effect, i.e, the accumulation of localized states on the system boundaries. Arguably, these nonhermitian effects are physical: The time dependence leads to local nonhermitian gain and loss processes on the lattice and nonunitary growth or decay of the time evolution of the field. Conversely, space dependence leads to the nonhermitian skin effect with spatial decay of the fields in a preferential direction. Hence, temporal curvature gradients produce nonunitary gain or loss, while spatial curvature gradients correspond to the nonhermitian skin effect. This suggests a duality between nonhermitian phenomena and spacetime deformations. This metric-induced nonhermiticity unveils an unexpected connection between the spacetime metric and nonhermitian phases of matter.

# 1   Introduction

Reconciling general relativity with quantum field theory in a mathematically and physically consistent way is an open problem in high-energy physics. However, a well-studied semiclassical approximation of quantum gravity is the quantum field theory in curved spacetime, which is obtained when particle fields are treated as quantum-mechanical, and spacetime is treated as a classical background [1, 2]. In this framework, e.g., the Dirac equation is extended from the flat Minkowski spacetime of quantum field theory to a curved spacetime by substituting the partial derivatives with covariant ones. Nevertheless, this theory itself is not free from apparent inconsistencies, such as the issue of the nonhermicity of the Dirac Hamiltonian in curved spacetime [3, 4]. This nonhermicity is usually cured by adding extra terms that make the Hamiltonian hermitian [4, 5], or by considering sufficiently smoothly varying metrics, which makes the nonhermitian terms negligible. The issue of hermiticity or nonhermiticity of the Dirac Hamiltonian is also relevant in light of the recent advances in the study of nonhermitian quantum mechanics [6, 7] and nonhermitian quantum field theories [8], and on the growing field of analog gravity, i.e., the study of quantum systems that simulate curved spacetime, such as Bose-Einstein condensates [9–12], optical metamaterials [13–17], cold atoms in optical lattices [18], graphene [19–25], and Weyl semimetals [26–29]. These systems are described by effective Hamiltonians corresponding to a Dirac equation regularized on a lattice and are usually derived perturbatively, specifically for the condensed matter system considered [19–25].

Here, I consider an alternative regularization scheme and derive an effective lattice Hamiltonian by regularizing the continuum Dirac equation in curved spacetime, regardless of the specifics of the condensed matter system considered. Generally, regularization on discrete lattices approximates the derivatives of the field with finite differences. However, this approach leads to nonhermitian lattice Hamiltonians, which can be made hermitian only by imposing hermiticity by hand [18]. Instead, here I first rewrite the Hamiltonian in terms of the derivatives of the field times a scaling function, which is related to the determinant of the metric, and then approximate these "renormalized" derivatives with finite differences. This regularization approach returns a hermitian or pseudohermitian Hamiltonian for time-independent (static) spacetime coordinates, without arbitrarily adding extra terms to cancel out the nonhermitian terms. Notably, I found that these pseudohermitian Hamiltonians have unbroken $\mathcal{PT}$-symmetry, corresponding to real energy spectra and unitary time evolutions. Conversely, for spacetime coordinates that depend explicitly on the time coordinate, I found that the lattice Hamiltonian is, in general, neither hermitian nor pseudohermitian: I argue that this nonhermicity is physical and that it corresponds to local gain and loss nonhermitian processes on the

lattice, resulting in the nonunitary time evolution of the field. On top of that, I found that for spacetime coordinates that depend explicitly on space, the lattice Hamiltonian exhibits the so-called nonhermitian skin effect [6,7] due to the presence of nonreciprocal couplings in the left and right directions: Consequently, the wavefunctions decay exponentially in a preferential direction and become localized at one of the boundaries when the Hamiltonian is regularized on a finite patch of the spacetime. I illustrate these general results by considering several examples, such as the Weyl [30], Rindler [31], de Sitter (dS), and anti-de Sitter (AdS) metrics [32,33]. Notably, I find that while the continuum theory is covariant under coordinate transformations, this covariance is lost upon lattice regularization: Consequently, some lattice properties, including pseudohermiticity, become coordinate dependent. Finally, I speculate on possible experimental realizations of quantum analogs of curved spacetime using condensed matter systems that simulate the lattice Hamiltonians derived here.

## 2 Dirac equation in curved 1+1D spacetime

Let us start with the Dirac equation in curved 1+1D spacetime [34–36]

$$\left[ i\gamma^a e_a{}^\mu \partial_\mu + \frac{i}{2}\gamma^a \frac{1}{\sqrt{-g}} \partial_\mu(\sqrt{-g}\, e_a{}^\mu) - M \right]\psi = 0, \tag{1}$$

where $\psi$ is a two-spinor, $\gamma^\mu$ the flat spacetime gamma matrices satisfying $\{\gamma^\mu, \gamma^\nu\} = 2\eta^{\mu\nu}$ with $\eta_{\mu\nu} = \mathrm{diag}(1,-1)$ the Minkowski metric, $\sqrt{-g}$ is the square root of the determinant of the metric, and the zweibein is related to the metric by the relations $g_{\mu\nu} = e^a{}_\mu e^b{}_\nu \eta_{ab}$, and $\eta_{ab} = e_a{}^\mu e_b{}^\nu g_{\mu\nu}$, with $g^{\mu\nu} = (g_{\mu\nu})^{-1}$. In the Weyl representation, $\gamma^0 = \sigma_x$ and $\gamma^1 = i\sigma_y$. The conserved scalar product in curved spacetime is defined as [3,4]

$$(\phi, \varphi) = -\int \mathrm{d}x\, \sqrt{-g}\, \phi^\dagger \gamma^0 e_a{}^0 \gamma^a \varphi, \tag{2}$$

which gives $(\phi, H\phi) - (H\phi, \phi) = i\int \mathrm{d}x\, \phi^\dagger \gamma^0 \partial_0(\sqrt{-g}\, e_a{}^0 \gamma^a)\phi$ (see Ref. [4]). This mandates that, for time-dependent metrics, the Hamiltonian is generally nonhermitian.

## 3 Conformally flat coordinates

Any metrics in 1+1D can be written in conformally flat coordinates as

$$\mathrm{d}s^2 = \Omega^2 \left( \mathrm{d}t^2 - \mathrm{d}x^2 \right), \tag{3}$$

with $\Omega \geq 0$ the conformal factor. Coordinate singularities correspond to points or regions in spacetime where $\Omega = 0$ or $\Omega = \infty$. The nonvanishing components of the the metric tensor are $g_{00} = \Omega^2$ and $g_{11} = -\Omega^2$, square root of the determinant $\sqrt{-g} = \Omega^2$, and nonzero zweibein $e_0{}^0 = \Omega^{-1}$ and $e_1{}^1 = \Omega^{-1}$. Separating time and space terms, the time evolution of the spinor field reads

$$i\partial_0 \psi = H\psi = \left( -i\gamma_0\gamma^1 \partial_1 - \frac{i}{2}\gamma_0\gamma^1 \frac{\partial_1 \Omega}{\Omega} - \frac{i}{2}\frac{\partial_0 \Omega}{\Omega} + M\gamma_0\Omega \right)\psi, \tag{4}$$

where $\gamma_0 = (\gamma^0)^{-1}$, and where the single-particle Hamiltonian density $H$ is identified as the operator acting on the spinor on the right side of the equation. This can be conveniently recast as

$$i\partial_0 \psi = H\psi = -i\gamma_0\gamma^1 \frac{\partial_1\left(\sqrt{\Omega}\psi\right)}{\sqrt{\Omega}} - \frac{i}{2}\frac{\partial_0\Omega}{\Omega}\psi + M\Omega\gamma_0\psi, \tag{5}$$

using $\partial_1(\sqrt{\Omega}) = \partial_1 \Omega / 2\sqrt{\Omega}$. For time-independent (static) coordinates $\partial_0 \Omega = 0$, this Hamiltonian is hermitian with respect to the conserved scalar product defined in Eq. (2), which in this case gives $(\phi, H\phi) - (H\phi, \phi) = 0$.

Single particle Hamiltonians can be discretized [37] on a lattice $x = na$ by substituting the spatial derivatives with finite differences $\partial_1 \psi \approx \frac{1}{2a}(\psi_{n+1} - \psi_{n-1})$. However, this approach leads to nonhermitian lattice Hamiltonians even for time-independent coordinates (see Appendix A), unless one imposes hermiticity by hand using $\psi^\dagger H \psi \to (H\psi)^\dagger \psi + \psi^\dagger H \psi$ [see, e.g., Eq. (15) in Ref. [18]]. Here, we can use more conveniently [38]

$$\partial_1 \left( \sqrt{\Omega} \psi \right) \approx \frac{1}{2a} \left( \sqrt{\Omega_{n+1}} \psi_{n+1} - \sqrt{\Omega_{n-1}} \psi_{n-1} \right), \tag{6}$$

with $\Omega_n = \Omega(na, t)$ and $\psi_n = \psi(na, t)$ in Eq. (5), yielding

$$H\psi_n = -\frac{i}{2a} \gamma_0 \gamma^1 \left( \sqrt{\frac{\Omega_{n+1}}{\Omega_n}} \psi_{n+1} - \sqrt{\frac{\Omega_{n-1}}{\Omega_n}} \psi_{n-1} \right) - \frac{i}{2} \frac{\partial_0 \Omega_n}{\Omega_n} \psi_n + M \Omega_n \gamma_0 \psi_n. \tag{7}$$

Hence, the Hamiltonian is $\mathcal{H} = \int dx \, \psi^\dagger H \psi \approx a \sum_n \psi_n^\dagger H \psi_n$, which reads

$$\mathcal{H} = a \sum_n -\frac{i}{2a} \left( \sqrt{\frac{\Omega_{n+1}}{\Omega_n}} \psi_n^\dagger \gamma_0 \gamma^1 \psi_{n+1} - \sqrt{\frac{\Omega_n}{\Omega_{n+1}}} \psi_{n+1}^\dagger \gamma_0 \gamma^1 \psi_n \right)$$
$$- \frac{i}{2} \frac{\partial_0 \Omega_n}{\Omega_n} \psi_n^\dagger \psi_n + M \Omega_n \psi_n^\dagger \gamma_0 \psi_n, \tag{8}$$

which is nonhermitian on the lattice (i.e., $\mathcal{H}^\dagger \neq \mathcal{H}$) for $\partial_0 \Omega \neq 0$ and $\partial_1 \Omega \neq 0$. For flat spacetime coordinates $\partial_0 \Omega = \partial_1 \Omega = 0$, the third term vanishes, while $\sqrt{\Omega_{n+1}/\Omega_n} = \sqrt{\Omega_n/\Omega_{n+1}} = 1$ in the first and second terms, so that the Hamiltonian becomes hermitian on the lattice (i.e., $\mathcal{H}^\dagger = \mathcal{H}$) since the factor $i\gamma_0 \gamma^1$ is anti-hermitian. In general, the lattice Hamiltonian can be compactly written as

$$\mathcal{H} = a \sum_n -i t_n^{\mathrm{LR}} \psi_n^\dagger \gamma_0 \gamma^1 \psi_{n+1} + i t_n^{\mathrm{RL}} \psi_{n+1}^\dagger \gamma_0 \gamma^1 \psi_n - i \delta_n \psi_n^\dagger \psi_n + \epsilon_n \psi_n^\dagger \gamma_0 \psi_n, \tag{9}$$

with

$$t_n^{\mathrm{LR}} = \frac{1}{2a} \sqrt{\frac{\Omega_{n+1}}{\Omega_n}} = \frac{1}{2a} e^{\frac{1}{2} \partial_1 \log \Omega_n}, \tag{10a}$$

$$t_n^{\mathrm{RL}} = \frac{1}{2a} \sqrt{\frac{\Omega_n}{\Omega_{n+1}}} = \frac{1}{2a} e^{-\frac{1}{2} \partial_1 \log \Omega_n}, \tag{10b}$$

$$\delta_n = \frac{1}{2} \frac{\partial_0 \Omega_n}{\Omega_n} = \frac{1}{2} \partial_0 \log \Omega_n, \tag{10c}$$

$$\epsilon_n = M \Omega_n, \tag{10d}$$

where $2a t_n^{\mathrm{LR}} = (2a t_n^{\mathrm{RL}})^{-1}$, and with $\partial_1 f_n = f_{n+1} - f_n$ with abuse of notation. The first two coefficients can be identified as the left-to-right and right-to-left "hopping" terms of the tight-binding model in Eq. (8) in the language of condensed matter physics. These hopping terms are not reciprocal, i.e., the left-to-right hoppings $n \to n+1$ are not equal to the right-to-left hoppings $n \to n-1$, as long as $\partial_1 \Omega \neq 0$. For space-independent coordinates $\partial_1 \Omega = 0$ the hopping terms become equal and $t_n^{\mathrm{LR}} = t_n^{\mathrm{RL}} = 1/2a$. Indeed, this Hamiltonian is a generalization of the Hatano-Nelson model [7, 39, 40] with space-dependent hopping terms and space-dependent on-site energies. Crucially, the nonreciprocity of the hoppings entails the nonhermiticity of the Hamiltonian. The third coefficient instead describes nonhermitian gain and loss processes on

Conformally flat coordinates / $\mathcal{P}$, $\mathcal{T}$ symmetries

| metric | | coefficients | | $\mathcal{PT}$ | $\mathcal{P}$ | $\mathcal{T}$ |
|---|---|---|---|---|---|---|
| $\partial_0\Omega = 0,$ | $\partial_1\Omega = 0,$ | $2at_n^{\mathrm{LR}} = 2at_n^{\mathrm{RL}} = 1,$ | $\delta_n = 0,$ | ✓ | ✓ | ✓ |
| $\partial_0\Omega = 0,$ | $\partial_1\Omega \neq 0,$ | | $\delta_n = 0,$ | ✓ | | |
| $\partial_0\Omega \neq 0,$ | $\partial_1\Omega = 0,$ | $2at_n^{\mathrm{LR}} = 2at_n^{\mathrm{RL}} = 1,$ | | | | ✓ |
| $\partial_0\Omega \neq 0,$ | $\partial_1\Omega \neq 0,$ | | | | | |

Table 1: Space-inversion and time-inversion symmetries of the lattice Hamiltonian and coefficients in Eq. (9) corresponding to conformally flat coordinates $\mathrm{d}s^2 = \Omega^2(\mathrm{d}t^2 - \mathrm{d}x^2)$ in 1+1D with conformal factor $\Omega$. Here, $\mathcal{P}$ represents the space-inversion symmetry (described by the unitary operator acting as $n + 1 \to n - 1$ on the lattice site indexes), $\mathcal{T}$ represents time-reversal symmetry (described by the antiunitary complex conjugation operator), and $\mathcal{PT}$ their composition. $\mathcal{T}$-symmetric Hamiltonians are hermitian, while $\mathcal{PT}$-symmetric Hamiltonians are pseudohermitian.

Conformally flat coordinates / TT, ST symmetries

| mass | metric | | coefficients | | TT | ST |
|---|---|---|---|---|---|---|
| $M \neq 0,$ | $\partial_0\Omega = 0,$ | $\partial_1\Omega = 0,$ | $2at_n^{\mathrm{LR}} = 2at_n^{\mathrm{RL}} = 1,$ | $\delta_n = 0,$ | ✓ | ✓ |
| $M \neq 0,$ | $\partial_0\Omega = 0,$ | $\partial_1\Omega \neq 0,$ | | $\delta_n = 0,$ | ✓ | |
| $M \neq 0,$ | $\partial_0\Omega \neq 0,$ | $\partial_1\Omega = 0,$ | $2at_n^{\mathrm{LR}} = 2at_n^{\mathrm{RL}} = 1,$ | | | ✓ |
| $M \neq 0,$ | $\partial_0\Omega \neq 0,$ | $\partial_1\Omega \neq 0,$ | | | | |
| $M = 0,$ | $\partial_0^2\log\Omega = \partial_0\partial_1\log\Omega = 0,$ | $\partial_1\partial_0\log\Omega = \partial_1^2\log\Omega = 0,$ | | $\epsilon_n = 0,$ | ✓ | ✓ |
| $M = 0,$ | $\partial_0^2\log\Omega = \partial_0\partial_1\log\Omega = 0,$ | otherwise | | $\epsilon_n = 0,$ | ✓ | |
| $M = 0,$ | otherwise | $\partial_1\partial_0\log\Omega = \partial_1^2\log\Omega = 0,$ | | $\epsilon_n = 0,$ | | ✓ |
| $M = 0,$ | otherwise | otherwise | | $\epsilon_n = 0,$ | | |

Table 2: Time-translation and space-translation symmetries of the lattice Hamiltonian with mass $M$ and coefficients in Eq. (9) corresponding to conformally flat coordinates $\mathrm{d}s^2 = \Omega^2(\mathrm{d}t^2 - \mathrm{d}x^2)$ in 1+1D with conformal factor $\Omega$. Here, TT represents time-translation symmetry, and ST represents space-translation symmetry. Note that $\partial_1 f_n = f_{n+1} - f_n$ with abuse of notation.

the lattice. For time-independent coordinates $\partial_0\Omega = 0$, the third term vanishes $\delta_n = 0$. For flat spacetime coordinates $\partial_0\Omega = \partial_1\Omega = 0$, then $t_n^{\mathrm{LR}} = t_n^{\mathrm{RL}} = 1/2a$ and $\delta_0 = 0$, so that the Hamiltonian becomes hermitian on the lattice (i.e., $\mathcal{H}^\dagger = \mathcal{H}$) since the factor $\mathrm{i}\gamma_0\gamma^1$ is anti-hermitian. The fourth coefficient corresponds to a time-dependent and spatially-dependent on-site potential energy on the discrete lattice, which renormalizes the mass term in Eq. (8). The presence of coordinate singularities leads to some of the coefficients diverging or vanishing somewhere on the lattice.

I will now briefly elucidate the symmetry properties of the lattice Hamiltonian in Eq. (8), in particular symmetry with respect to time translations, space translations, time reversal, and space inversion. The lattice Hamiltonian in Eq. (8) is time-independent (invariant up to time translations) for time-independent coordinates $\partial_0\Omega = 0$, and for time-dependent coordinates $\partial_0\Omega \neq 0$ in the massless case $M = 0$ as long as $\partial_0\log\Omega_n$ and $\partial_1\log\Omega_n$ are time-independent, i.e., as long as $\partial_0^2\log\Omega = 0$ and $\partial_0\partial_1\log\Omega = 0$. Similarly, the lattice Hamiltonian is translational invariant (invariant up to space-translations) for space-independent coordinates $\partial_1\Omega = 0$, and for space-dependent coordinates $\partial_1\Omega \neq 0$ in the massless case $M = 0$ as long as $\partial_0\log\Omega_n$ and $\partial_1\log\Omega_n$ are space-independent, i.e., as long as $\partial_1\partial_0\log\Omega = 0$ and $\partial_1^2\log\Omega = 0$.

Space-inversion symmetry is described by the space-inversion symmetry operator $\mathcal{P}$, de-

Conformally flat coordinates / Examples

| metric | coefficients | | | | TT | ST | $\mathcal{PT}$ | $\mathcal{P}$ | $\mathcal{T}$ |
|---|---|---|---|---|---|---|---|---|---|
| Weyl $\Omega = e^{rt+qx}$ | $2at_n^{\mathrm{LR}} = (2at_n^{\mathrm{RL}})^{-1} = e^{\frac{1}{2}qa},$ | $\delta_n = \frac{r}{2},$ | $\epsilon_n = Me^{rt+qna}$ | | if $M=0$ | if $M=0$ | | | |
| Rindler $\Omega = e^{qx}$ | $2at_n^{\mathrm{LR}} = (2at_n^{\mathrm{RL}})^{-1} = e^{\frac{1}{2}qa},$ | $\delta_n = 0,$ | $\epsilon_n = Me^{qna}$ | | $\checkmark$ | if $M=0$ | $\checkmark$ | | |
| $\Omega = e^{rt}$ | $2at_n^{\mathrm{LR}} = 2at_n^{\mathrm{RL}} = 1,$ | $\delta_n = \frac{r}{2},$ | $\epsilon_n = Me^{rt}$ | | if $M=0$ | $\checkmark$ | | $\checkmark$ | |
| $\Omega = (rt+qx)^{-1}$ | $2at_n^{\mathrm{LR}} = (2at_n^{\mathrm{RL}})^{-1} = \sqrt{\frac{rt+qna}{rt+qna+qa}},$ | $\delta_n = \frac{r}{2(rt+qna)},$ | $\epsilon_n = \frac{M}{rt+qna}$ | | | | | | |
| AdS $\Omega = (qx)^{-1}$ | $2at_n^{\mathrm{LR}} = (2at_n^{\mathrm{RL}})^{-1} = \sqrt{\frac{n}{n+1}},$ | $\delta_n = 0,$ | $\epsilon_n = \frac{M}{qna}$ | | $\checkmark$ | | $\checkmark$ | | |
| dS $\Omega = (rt)^{-1}$ | $2at_n^{\mathrm{LR}} = 2at_n^{\mathrm{RL}} = 1,$ | $\delta_n = -\frac{1}{2t},$ | $\epsilon_n = \frac{M}{rt}$ | | | $\checkmark$ | | $\checkmark$ | |

Table 3: Examples of conformally flat coordinates $\mathrm{d}s^2 = \Omega^2(\mathrm{d}t^2 - \mathrm{d}x^2)$ and coefficients of their corresponding lattice Hamiltonian in Eq. (9), including the Weyl [30], Rindler [31], de Sitter (dS), and anti-de Sitter (AdS) metrics [32,33]. In the case of the time-dependent Weyl metric, one obtains a lattice Hamiltonian that is not pseudohermitian, and that is moreover time-dependent and space-dependent for $M \neq 0$ but time-independent with space-independent (having time-translational and space-translational symmetries) for $M = 0$. The lattice Hamiltonian in the Rindler metric is instead pseudohermitian, and exhibits time-translational but no space-translational symmetry, with the exception of the massless case $M = 0$. The lattice Hamiltonian in the anti-de Sitter metric (defined on the patch $x > 0$) is pseudohermitian, and exhibits time-translational but no space-translational symmetry. The lattice Hamiltonian in the de Sitter metric (defined on the patch $t > 0$) is not pseudohermitian, and exhibits space-translational and space-inversion symmetry but no time-translational symmetry. The metric on the fourth row is a generalization of both the de Sitter and anti-de Sitter metrics, and it is defined on the patch $x, t > 0$.

fined as the unitary operator acting as $n+1 \to n-1$ on the lattice site indexes. Space-inversion symmetry requires that $t_n^{\mathrm{RL}} = t_n^{\mathrm{LR}}$, i.e., it requires space-independent coordinates. Note that in the space-independent case, the lattice Hamiltonian is not only translational invariant but also invariant with respect to space inversion $\mathcal{P}$. However, translational invariance alone does not mandate space inversion symmetry, such in the massless case $M = 0$ where $\Omega_{n+1}/\Omega_n$ and $(\partial_0 \Omega_n)/\Omega_n$ are space-independent and $\Omega_{n+1}/\Omega_n > 1$.

Time-reversal symmetry is described by the time-reversal symmetry operator $\mathcal{T}$, defined as the antiunitary complex conjugation operator. Time-reversal symmetry thus coincides with the property of being hermitian. Time-reversal symmetry requires that $t_n^{\mathrm{RL}} = t_n^{\mathrm{LR}}$ and $\delta_n = 0$, i.e., it requires flat coordinates. Although not hermitian in general, for time-independent coordinates $\partial_0 \Omega = 0$ (which gives $\delta_n = 0$), the lattice Hamiltonian in Eq. (8) is pseudohermitian and $\mathcal{PT}$-symmetric [41–43], i.e., $\mathcal{H} = \mathcal{PTH}(\mathcal{PT})^{-1} = \mathcal{PH}^*\mathcal{P}$, with $\mathcal{P}$ the space-inversion and $\mathcal{T}$ the time-reversal symmetry operators. In this case, using the similarity transformation

$$\psi_n \to \frac{1}{\sqrt{\Omega_n}}\psi_n = e^{\mathrm{i}\theta_n}\psi_n, \quad \psi_n^\dagger \to \sqrt{\Omega_n}\psi_n^\dagger = e^{-\mathrm{i}\theta_n}\psi_n^\dagger, \tag{11}$$

which can be seen as a gauge transformation with imaginary angles [7,39,40] $\theta_n = \frac{\mathrm{i}}{2}\log\Omega_n$, returns the hermitian Hamiltonian

$$\mathcal{H}' = a\sum_n -\frac{\mathrm{i}}{2a}\left(\psi_n^\dagger\gamma_0\gamma^1\psi_{n+1} - \psi_{n+1}^\dagger\gamma_0\gamma^1\psi_n\right) + M\Omega_n\psi_n^\dagger\gamma_0\psi_n, \tag{12}$$

which is not unitarily equivalent, but only isospectral to Eq. (8). Note that this transformation makes the hopping term reciprocal $t_n' = \sqrt{t_n^{\mathrm{LR}}t_n^{\mathrm{RL}}} = 1/2a$ and leaves the on-site energy term unchanged. The Hamiltonian $\mathcal{H}'$ is now invariant with respect to space-inversion $\mathcal{P}$ and time-inversion $\mathcal{T}$ separately. Moreover, it is translationally invariant for space-independent coordinates $\partial_1 \Omega = 0$ and for space-dependent coordinates $\partial_1 \Omega \neq 0$ in the massless case $M = 0$.

Consequently, the Hamiltonian $\mathcal{H}$ in Eq. (8) for time-independent coordinates $\partial_0 \Omega = 0$ has unbroken $\mathcal{PT}$-symmetry, having real energy spectra corresponding to unitary time evolution. Hence, the nonreciprocity of the hoppings entails the nonhermicity of the Hamiltonian, but preserves the pseudohermicity and does not break the $\mathcal{PT}$-symmetry. On the other hand, time-dependent coordinates break pseudohermicity and $\mathcal{PT}$-symmetry.

The properties of lattice Hamiltonians corresponding to 1+1D metrics in conformally flat coordinates are summarized in Tables 1 and 2. Some examples, including the Rindler, de Sitter, and anti-de Sitter metrics are listed in Table 3.

Notice that in the Rindler metric in the massless case $M = 0$, the lattice Hamiltonian coincides with that of the original Hatano-Nelson model [39,40,44] (without disorder), which is the archetype of nonhermitian models in condensed matter physics. Indeed, applying the imaginary gauge transformation in Eq. (22), which becomes $\psi_n \to e^{-\frac{1}{2}qna}\psi_n$, $\psi_n^\dagger \to e^{\frac{1}{2}qna}\psi_n^\dagger$, returns simply $\widetilde{\mathcal{H}} = a \sum_n -\frac{i}{2a} \left( \psi_n^\dagger \gamma_0 \gamma^1 \psi_{n+1} - \psi_{n+1}^\dagger \gamma_0 \gamma^1 \psi_n \right)$.

Discrete tight-binding Hamiltonians can be simulated in several different condensed matter systems, arrays of atoms [45, 46], quantum dots [47], and cold atoms in optical lattices [18, 48–50]. A common approach to probe their quantum state is to measure the electronic or atomic density of states, i.e., the density of energy levels (as a function of energy). In some systems, one can also measure the electronic or atomic *local* density of states, which is the spatially resolved density of energy levels (as a function of energy and position). This is a direct probe of the wavefunction of the electrons or atoms in the condensed matter system. Hence, to relate the theoretical results obtained here to experimentally accessible physical properties, I show below the local density of states of the lattice Hamiltonians considered here, calculated following the method detailed in Appendix B. Figure 1 shows the local density of states of lattice Hamiltonians corresponding to the Dirac equation in curved spacetimes with conformally flat coordinates in the Rindler, de Sitter, and anti-de Sitter metrics in Table 3 as a function of energy and position in the massless and massive cases. The lattice Hamiltonian is pseudohermitian for the Rindler and anti-de Sitter metrics. Instead, the lattice Hamiltonian is not pseudohermitian for the de Sitter metric and exhibits complex energy eigenvalues which are time-dependent (not shown). Figure 2 shows the local density of states on the real and imaginary axes of lattice Hamiltonians corresponding to the Dirac equation in curved spacetime with a conformally flat metric with coordinates $ds^2 = (rt + qx)^{-1}(dt^2 - dx^2)$ [see Table 3] as a function of energy and position and at different time slices in the massless and massive cases. This lattice Hamiltonian is not hermitian and not pseudohermitian, and it is time-dependent. The density of states and the energy spectra are gapless in the massless case, while they become gapped in the massive case.

# 4   Diagonal coordinates

As stated before, all metrics in 1+1D spacetime can be recast in a conformally flat form as in Eq. (3). Obviously, the geometrical structure of spacetime and all physical observables are invariant upon a change of coordinates, as a consequence of general covariance. However, when regularizing the spatial dimension on the lattice, this invariance is broken. Indeed, the discrete lattice introduces a preferred spatial frame and an artificial distinction between space and time. As a result, some physical properties can become dependent on the specific choice of the coordinates. To show how this can happen, in this Section, I will consider coordinates corresponding to a diagonal metric tensor. Another reason to consider these more general forms of the metric is that some of the results obtained in this Section may be easier to generalize to higher dimensions, in particular in 2+1D and 3+1D spacetimes, where spacetimes are not necessarily conformally flat.

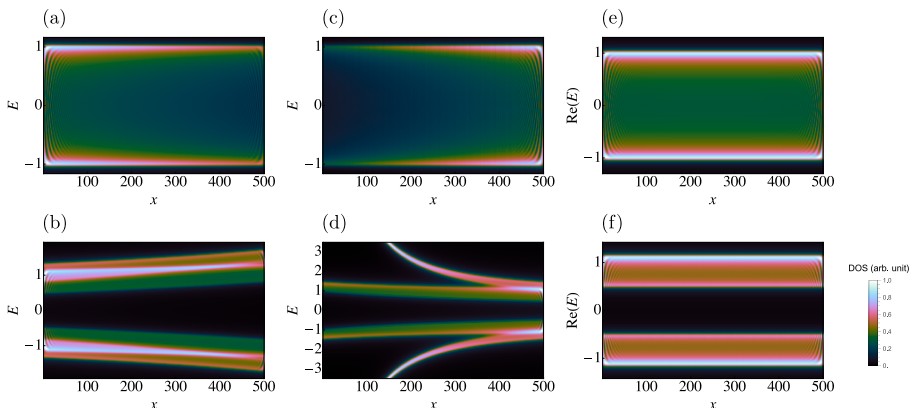

Figure 1: Local density of states (LDOS) of the lattice Hamiltonians corresponding to the Dirac equation in curved spacetimes with conformally flat coordinates [see Table 3], calculated on a finite patch as a function of the energy and position. Different panels correspond to: Rindler metric in conformally flat coordinates $ds^2 = e^{qx}(dt^2 - dx^2)$ in the massless (a) and massive (b) cases; anti-de Sitter metric in conformally flat coordinates $ds^2 = (qx)^{-1}(dt^2 - dx^2)$ with coordinate singularity at $x = 0$ in the massless (c) and massive (d) cases; de Sitter metric in conformally flat coordinates $ds^2 = (rt)^{-1}(dt^2 - dx^2)$ with coordinate singularity at $t = 0$ in the massless (e) and massive (f) cases; In the de Sitter case, the lattice Hamiltonian has no $\mathcal{PT}$ symmetry and thus exhibits complex energy eigenvalues, with the imaginary part of the LDOS (not shown) being time-dependent, while the real part of the LDOS remains time-independent. The mass in (b), (d), and (f) is $M = 0.5$.

190 Hence, we consider a general metric in the form

$$ds^2 = \alpha^2 dt^2 - \beta^2 dx^2, \tag{13}$$

191 with $\alpha, \beta \geq 0$, yielding the metric tensor $g_{00} = \alpha^2$ and $g_{11} = -\beta^2$, square root of the determi-
192 nant $\sqrt{-g} = \alpha\beta$, and zweibein $e_0{}^0 = \alpha^{-1}$ and $e_1{}^1 = \beta^{-1}$. Coordinate singularities correspond
193 to points or regions in spacetime where $\alpha\beta = 0$ or $\alpha\beta = \infty$. Separating time and space terms,
194 this metric yields

$$i\partial_0 \psi = H\psi = \left( -i\gamma_0\gamma^1 \frac{\alpha}{\beta}\partial_1 - \frac{i}{2}\frac{\partial_0\beta}{\beta} - \frac{i}{2}\gamma_0\gamma^1 \frac{\partial_1\alpha}{\beta} + M\gamma_0\alpha \right)\psi, \tag{14}$$

195 which can conveniently be recast as

$$i\partial_0 \psi = H\psi = -\frac{i\sqrt{\alpha}}{\beta}\gamma_0\gamma^1\partial_1\left(\sqrt{\alpha}\psi\right) - \frac{i}{2}\frac{\partial_0\beta}{\beta}\psi + M\alpha\gamma_0\psi, \tag{15}$$

196 which is again hermitian for time-independent coordinates. Regularizing on the lattice using
197 Eq. (6) for $\sqrt{\alpha}\psi$ yields

$$H\psi_n = -\frac{i}{2a}\gamma_0\gamma^1 \frac{\sqrt{\alpha_n}}{\beta_n}\left(\sqrt{\alpha_{n+1}}\psi_{n+1} - \sqrt{\alpha_{n-1}}\psi_{n-1}\right) - \frac{i}{2}\frac{\partial_0\beta_n}{\beta_n}\psi_n + M\alpha_n\gamma_0\psi_n, \tag{16}$$

198 with $\alpha_n = \alpha(na, t)$, $\beta_n = \beta(na, t)$, and $\psi_n = \psi(na, t)$. Hence, the Hamiltonian is

$$\mathcal{H} = a\sum_n -\frac{i}{2a}\left( \frac{\sqrt{\alpha_n\alpha_{n+1}}}{\beta_n}\psi_n^\dagger\gamma_0\gamma^1\psi_{n+1} - \frac{\sqrt{\alpha_n\alpha_{n+1}}}{\beta_{n+1}}\psi_{n+1}^\dagger\gamma_0\gamma^1\psi_n \right)$$
$$-\frac{i}{2}\frac{\partial_0\beta_n}{\beta_n}\psi_n^\dagger\psi_n + M\alpha_n\psi_n^\dagger\gamma_0\psi_n, \tag{17}$$

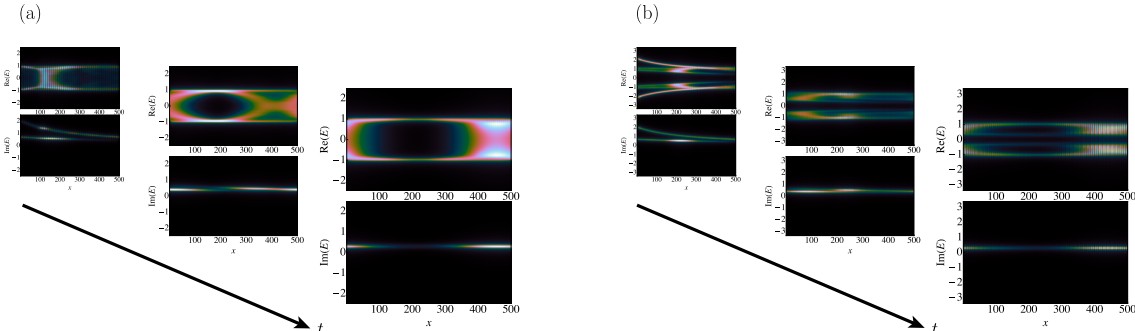

Figure 2: Local density of states (LDOS) on the real and imaginary axes of the time-dependent and nonhermitian lattice Hamiltonian corresponding to curved spacetime with the conformally flat metric $ds^2 = (rt + qx)^{-1}(dt^2 - dx^2)$ [see Table 3], calculated on a finite patch as a function of the energy and position and at different time slices $t = 0.25, 0.5, 0.75$ with $r = 1$. For time-dependent Hamiltonians, the time-dependent local density of states shown here describes the adiabatic time-evolution of the energy spectra. Different panels correspond to the massless (a) and massive (b) cases ($M = 0.5$). This metric has a coordinate singularity at $x = t = 0$.

which is nonhermitian on the lattice for $\partial_0 \beta \neq 0$ and $\partial_1 \beta \neq 0$. Also in this case, the Hamiltonian can be written in compact form as in Eq. (9) with

$$t_n^{\text{LR}} = \frac{1}{2a} \frac{\sqrt{\alpha_n \alpha_{n+1}}}{\beta_n} = \frac{1}{2a} e^{\frac{1}{2} \log(\alpha_n \alpha_{n+1}) - \log \beta_n}, \tag{18a}$$

$$t_n^{\text{RL}} = \frac{1}{2a} \frac{\sqrt{\alpha_n \alpha_{n+1}}}{\beta_{n+1}} = \frac{1}{2a} e^{\frac{1}{2} \log(\alpha_n \alpha_{n+1}) - \log \beta_{n+1}}, \tag{18b}$$

$$\delta_n = \frac{1}{2} \frac{\partial_0 \beta_n}{\beta_n} = \frac{1}{2} \partial_0 \log \beta_n, \tag{18c}$$

$$\epsilon_n = M \alpha_n, \tag{18d}$$

where the first two coefficients are the left-to-right and right-to-left hoppings, the third coefficient again describes nonhermitian gain and loss processes, and the fourth term renormalizes the mass term as a time-dependent and spatially-dependent on-site potential energy. The hopping terms are not reciprocal as long as $\partial_1 \beta \neq 0$, and become equal $t_n^{\text{LR}} = t_n^{\text{RL}} = (1/2a)\sqrt{\alpha_n \alpha_{n+1}}/\beta_n$ for $\partial_1 \beta = 0$. The third term vanishes $\delta_n = 0$ if $\partial_0 \beta = 0$. Again, the presence of coordinate singularities leads to some of the coefficients diverging or vanishing somewhere on the lattice. These coefficients reduce to Eq. (20) for $\alpha = \beta = \Omega$.

I will now briefly elucidate the symmetry properties of the Hamiltonian in Eq. (17), similarly to the previous Section. The lattice Hamiltonian is time-independent (invariant up to time translations) for time-independent coordinates $\partial_0 \alpha = \partial_0 \beta = 0$, and for time-dependent coordinates in the massless case $M = 0$ as long as $\partial_0 \log \beta_n$, $\log(\alpha_n \alpha_{n+1}) - 2 \log \beta_n$, and $\log(\alpha_n \alpha_{n+1}) - 2 \log \beta_{n+1}$ are time-independent, i.e., as long as $\partial_0^2 \log \beta = 0$, $\partial_0 \partial_1 \log \beta = 0$, and $\partial_0 \partial_1 \log(\alpha_n \alpha_{n+1}) = 0$. Similarly, the lattice Hamiltonian is translational invariant (invariant up to space translations) for space-independent coordinates $\partial_1 \alpha = \partial_1 \beta = 0$, and for space-dependent coordinates in the massless case $M = 0$ as long as $\partial_0 \log \beta_n$, $\log(\alpha_n \alpha_{n+1}) - 2 \log \beta_n$, and $\log(\alpha_n \alpha_{n+1}) - 2 \log \beta_{n+1}$ are space-independent, i.e., as long as $\partial_1 \partial_0 \log \beta = 0$, $\partial_1^2 \log \beta = 0$, and $\partial_1^2 \log(\alpha_n \alpha_{n+1}) = 0$ [1]. Space-inversion symmetry $\mathcal{P}$ requires that $t_n^{\text{RL}} = t_n^{\text{LR}}$, i.e., $\partial_1 \beta = 0$.

---

[1] If $\log(\alpha_n \alpha_{n+1}) - 2 \log \beta_n$, and $\log(\alpha_n \alpha_{n+1}) - 2 \log \beta_{n+1}$ are space-independent (or time-independent), then their difference $\log \beta_{n+1} - \log \beta_n = \partial_1 \beta_n$ must be space-independent (or time-independent). Moreover, under the same assumptions $\log(\alpha_{n+1} \alpha_{n+2}) - 2 \log \beta_{n+1}$ is space-independent (or time-independent), then $\log(\alpha_{n+1} \alpha_{n+2}) - \log(\alpha_n \alpha_{n+1}) = \partial_1 \log(\alpha_n \alpha_{n+1})$ must be space-independent (or time-independent).

Diagonal coordinates / $\mathcal{P}$, $\mathcal{T}$ symmetries

| metric | | coefficients | | $\mathcal{P}\mathcal{T}$ | $\mathcal{P}$ | $\mathcal{T}$ |
|---|---|---|---|---|---|---|
| $\partial_0\beta=0$, | $\partial_1\beta=0$, | $2at_n^{\mathrm{LR}}=2at_n^{\mathrm{RL}}=\frac{\sqrt{\alpha_n\alpha_{n+1}}}{\beta_1}$, | $\delta_n=0$, | ✓ | ✓ | ✓ |
| $\partial_0\beta=0$, | $\partial_1\beta\neq0$, | | $\delta_n=0$, | ✓ | | |
| $\partial_0\beta\neq0$, | $\partial_1\beta=0$, | $2at_n^{\mathrm{LR}}=2at_n^{\mathrm{RL}}=\frac{\sqrt{\alpha_n\alpha_{n+1}}}{\beta_1}$, | | | | ✓ |
| $\partial_0\beta\neq0$, | $\partial_1\beta\neq0$, | | | | | |

Table 4: Space-inversion and time-inversion symmetries of the lattice Hamiltonian and coefficients in Eq. (9) corresponding to diagonal coordinates $\mathrm{d}s^2=\alpha^2\mathrm{d}t^2-\beta^2\mathrm{d}x^2$ in 1+1D.

Diagonal coordinates / TT, ST symmetries

| mass | metric | | | coefficients | | TT | ST |
|---|---|---|---|---|---|---|---|
| $M\neq0$, | $\partial_0\alpha=\partial_0\beta=0$, | $\partial_1\alpha=\partial_1\beta=0$, | | $2at_n^{\mathrm{LR}}=\frac{1}{2at_n^{\mathrm{RL}}}=\frac{\alpha_1}{\beta_1}$, | $\delta_n=0$, | ✓ | ✓ |
| $M\neq0$, | $\partial_0\alpha=\partial_0\beta=0$, | $\partial_1\alpha\neq0$ or $\partial_1\beta\neq0$, | | | $\delta_n=0$, | ✓ | |
| $M\neq0$, | $\partial_0\alpha\neq0$ or $\partial_0\beta\neq0$, | $\partial_1\alpha=\partial_1\beta=0$, | | $2at_n^{\mathrm{LR}}=\frac{1}{2at_n^{\mathrm{RL}}}=\frac{\alpha_1}{\beta_1}$, | | | ✓ |
| $M\neq0$, | $\partial_0\alpha\neq0$ or $\partial_0\beta\neq0$, | $\partial_1\alpha\neq0$ or $\partial_1\beta\neq0$, | | | | | |
| $M=0$, | $\partial_0^2\log\beta=\partial_0\partial_1\log\beta=\partial_0\partial_1\log(\alpha_n\alpha_{n+1})=0$, | $\partial_1\partial_0\log\beta=\partial_1^2\log\beta=\partial_1^2\log(\alpha_n\alpha_{n+1})=0$, | | | $\epsilon_n=0$, | ✓ | ✓ |
| $M=0$, | $\partial_0^2\log\beta=\partial_0\partial_1\log\beta=\partial_0\partial_1\log(\alpha_n\alpha_{n+1})=0$, | otherwise | | | $\epsilon_n=0$, | ✓ | |
| $M=0$, | otherwise | $\partial_1\partial_0\log\beta=\partial_1^2\log\beta=\partial_1^2\log(\alpha_n\alpha_{n+1})=0$, | | | $\epsilon_n=0$, | | ✓ |
| $M=0$, | otherwise | otherwise | | | $\epsilon_n=0$, | | |

Table 5: Time-translation and space-translation symmetries of the lattice Hamiltonian with mass $M$ and coefficients in Eq. (9) corresponding to diagonal coordinates $\mathrm{d}s^2=\alpha^2\mathrm{d}t^2-\beta^2\mathrm{d}x^2$ in 1+1D.

Conversely, time-reversal symmetry $\mathcal{T}$ requires that $t_n^{\mathrm{RL}}=t_n^{\mathrm{LR}}$ and $\delta_n=0$, i.e., $\partial_0\beta=\partial_1\beta=0$. For $\partial_0\beta=\partial_1\beta=0$ the Hamiltonian becomes

$$\mathcal{H}=a\sum_n-\frac{\mathrm{i}}{2a}\frac{\sqrt{\alpha_n\alpha_{n+1}}}{\beta_1}\left(\psi_n^\dagger\gamma_0\gamma^1\psi_{n+1}-\psi_{n+1}^\dagger\gamma_0\gamma^1\psi_n\right)+M\alpha_n\psi_n^\dagger\gamma_0\psi_n, \qquad (19)$$

which is hermitian, even when the metric coordinates are time-dependent $\partial_0\alpha\neq0$. We stress the fact that in the case $\partial_0\beta=\partial_1\beta=0$, the regularization approach in Eq. (6) leads to a hermitian lattice Hamiltonian in this case, while the more standard regularization (finite differences of the wavefunction) leads to nonhermitian lattice Hamiltonians even for time-independent coordinates unless $\partial_0\beta=\partial_1\beta=0$ and $\partial_1\alpha=0$, as shown in Appendix A. Moreover, the Hamiltonian exhibits both time-inversion and space-inversion symmetry since

$$t_n^{\mathrm{LR}}=t_n^{\mathrm{RL}}=\frac{1}{2a}\frac{\sqrt{\alpha_n\alpha_{n+1}}}{\beta_1}, \qquad (20)$$

which can be identified as the hopping term of a tight-binding model and is proportional to the geometric average of the square root of the determinant of the metric on two contiguous lattice sites. Introducing

$$\alpha=e^{-d(x)}, \quad \alpha_n=e^{-d_n}, \qquad (21)$$

Diagonal coordinates / Examples

| metric | coefficients | | | TT | ST | $\mathcal{PT}$ | $\mathcal{P}$ | $\mathcal{T}$ |
|---|---|---|---|---|---|---|---|---|
| Rindler $\alpha = qx,\ \beta = 1$ | $2at_n^{\mathrm{LR}} = 2at_n^{\mathrm{RL}} = qa\sqrt{n(n+1)},$ | $\delta_n = 0,$ | $\epsilon_n = Mqna$ | ✓ | | ✓ | ✓ | ✓ |
| $\alpha = rt,\ \beta = 1$ | $2at_n^{\mathrm{LR}} = 2at_n^{\mathrm{RL}} = rt,$ | $\delta_n = 0,$ | $\epsilon_n = Mrt$ | | ✓ | ✓ | ✓ | ✓ |
| AdS $\alpha = \frac{1}{\beta} = \sqrt{1+(qx)^2}$ | $2at_n^{\mathrm{LR}} = \sqrt[4]{[1+(qna)^2]^3[1+(qna+qa)^2]},$ | $\delta_n = 0,$ | $\epsilon_n = M\sqrt{1+(qna)^2}$ | ✓ | ✓ | | | |
| | $2at_n^{\mathrm{RL}} = \sqrt[4]{[1+(qna)^2][1+(qna+qa)^2]^3},$ | | | | | | | |
| dS $\alpha = \frac{1}{\beta} = \sqrt{1-(qx)^2}$ | $2at_n^{\mathrm{LR}} = \sqrt[4]{[1-(qna)^2]^3[1-(qna+qa)^2]},$ | $\delta_n = 0,$ | $\epsilon_n = M\sqrt{1-(qna)^2}$ | ✓ | ✓ | | | |
| | $2at_n^{\mathrm{RL}} = \sqrt[4]{[1-(qna)^2][1-(qna+qa)^2]^3},$ | | | | | | | |

Table 6: Examples of diagonal metric coordinates $\mathrm{d}s^2 = \alpha^2\mathrm{d}t^2 - \beta^2\mathrm{d}x^2$ and coefficients of their corresponding lattice Hamiltonian in Eq. (9), including the Rindler [31], de Sitter, and anti-de Sitter metrics [32, 33]. Notice that the lattice Hamiltonians considered here are all pseudohermitian and time-independent.

it is natural to interpret the quantity $(d_n + d_{n+1})/2$ as the distance (in some characteristic units) between the lattice sites $n$ and $n + 1$, at least on patches of the spacetime where $d(x) = -\log(\alpha) \geq 0$. Generally, hopping amplitudes in condensed matter describe overlap integrals between modes localized on a single lattice site (such as Wannier functions). These overlap integrals typically scale exponentially with the distance as $t \propto e^{-d_n/l}$, where $d_n$ is the distance between the modes and $l$ is a characteristic length scale describing the localization of the modes (e.g., end modes in topological insulators or superconductors or atomic modes in optical lattices). In these systems, the coefficient $\epsilon_n = M\alpha_n$ corresponds to a spatially-dependent (and possibly time-dependent) on-site energy potential. In this case, coordinate singularities $\alpha = 0$ have a straightforward physical interpretation on the lattice: The hopping coefficient vanishes while the distance $d_n$ diverges at the singularity. Conversely, for coordinate singularities $\alpha = \infty$, the hopping coefficient diverges at the singularity.

For time-independent coordinates and for time-dependent coordinates with $\partial_0\beta = 0$ (but not necessarily space-independent), the lattice Hamiltonian in Eq. (17) is pseudohermitian with unbroken $\mathcal{PT}$-symmetry, real spectra, and unitary time-evolution. Indeed in this case, using the similarity transformation

$$\psi_n \to \frac{1}{\sqrt{\beta_n}}\psi_n = e^{\mathrm{i}\theta_n}\psi_n, \quad \psi_n^\dagger \to \sqrt{\beta_n}\psi_n^\dagger = e^{-\mathrm{i}\theta_n}\psi_n^\dagger, \tag{22}$$

that is, a gauge transformation with imaginary angles $\theta_n = \frac{\mathrm{i}}{2}\log\beta_n$, returns the hermitian Hamiltonian

$$\mathcal{H}' = a\sum_n -\frac{\mathrm{i}}{2a}\sqrt{\frac{\alpha_n\alpha_{n+1}}{\beta_n\beta_{n+1}}}\left(\psi_n^\dagger\gamma_0\gamma^1\psi_{n+1} - \psi_{n+1}^\dagger\gamma_0\gamma^1\psi_n\right) + M\alpha_n\psi_n^\dagger\gamma_0\psi_n, \tag{23}$$

which is not unitarily equivalent but isospectral to Eq. (17) if $\partial_0\beta_n = 0$. As in the previous case, this transformation makes the hopping term reciprocal $t_n' = (1/2a)\sqrt{\alpha_n\alpha_{n+1}/\beta_n\beta_{n+1}}$ and leaves the on-site energy term unchanged. The Hamiltonian $\mathcal{H}'$ is invariant with respect to space-inversion $\mathcal{P}$ and time-inversion $\mathcal{T}$ separately. Moreover, it is translationally invariant for space-independent coordinates $\partial_1\alpha = \partial_1\beta = 0$ and for space-dependent coordinates in the massless case $M = 0$ as long as $\partial_1(\alpha_n/\beta_n) = 0$. Note that the condition for pseudohermicity in the case of diagonal metric tensors is weaker compared to the case of conformally flat coordinates.

The properties of lattice Hamiltonians corresponding to 1+1D diagonal metric tensor are summarized in Tables 4 and 5. Some examples are listed in Table 6. Notice that the lattice Hamiltonians corresponding to the Rindler and the anti-de Sitter spacetimes are time-independent and pseudohermitian in the conformally flat coordinates in Table 3 and in the

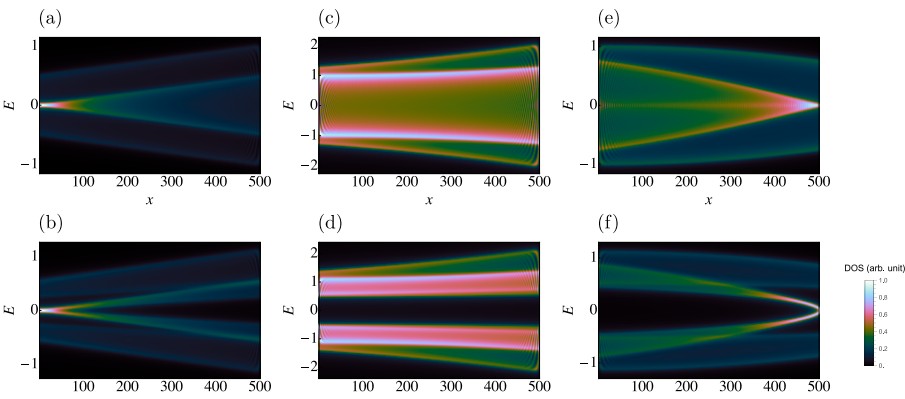

Figure 3: Local density of states (LDOS) of the lattice Hamiltonians corresponding to the Dirac equation in curved and spacetimes with diagonal metric tensors [see Table 6], calculated on a finite patch as a function of the energy and position. Different panels correspond to: Rindler metric $ds^2 = (qx)^2 dt^2 - dx^2$ with a coordinate singularity at $x = 0$ in the massless (a) and massive (b) cases; anti-de Sitter metric $ds^2 = \alpha^2 dt^2 - \beta^2 dx^2$ with $\alpha = 1/\beta = \sqrt{1 + (qx)^2}$ in the massless (c) and massive (d) cases; de Sitter metric $ds^2 = \alpha^2 dt^2 - \beta^2 dx^2$ with $\alpha = 1/\beta = \sqrt{1 - (qx)^2}$ with a coordinate singularity at $x = 1/q = N$ in the massless (e) and massive (f) cases. The mass in (b), (d), and (f) is $M = 0.5$.

diagonal coordinates chosen in Table 6. In particular, the lattice Hamiltonian for the Rindler metric is hermitian in diagonal coordinates, but only pseudohermitian in the conformally flat case. However, the lattice Hamiltonian corresponding to the de Sitter spacetime is not time-independent and not pseudohermitian in the conformally flat case, while it is time-independent and pseudohermitian in the diagonal coordinates chosen in Table 6. This is not surprising because, as anticipated, general covariance does not necessarily hold when the Dirac equation is regularized on the lattice. As a consequence, some physical properties, such as pseudohermicity, may depend on the choice of the coordinates.

Figure 3 shows the local density of states of lattice Hamiltonians corresponding to the Dirac equation in curved spacetimes with diagonal coordinates in the Rindler, de Sitter, and anti-de Sitter spacetimes in Table 6 as a function of energy and position in the massless and massive cases.

The Rindler and de Sitter metrics exhibit coordinate singularities at $x = 0$ and $x = 1/q$, respectively. These singularities correspond to localized zero-energy modes of the lattice Hamiltonian. The presence of a singularity in the Rindler and de Sitter metrics and the corresponding zero-energy mode mandates that the energy spectra and total density of states are always gapless (regardless of the mass term). In particular, the local density of states is gapless everywhere in the massless case, while it becomes locally gapped in the massive case, with the exception of the singularity at $x = 0$ (Rindler metric) and $x = 1/q$ (de Sitter metric). Remarkably, the Rindler metric approximates the Schwarzschild metric near a black hole horizon. For the anti-de Sitter metric instead (which has no coordinate singularity), the density of states and the energy spectra are gapless in the massless case, while they become gapped in the massive case, as expected.

## 5 Time evolution, skin effect, and nonhermiticity

Finally, we want to unveil the physical meaning of the nonunitary time evolution ensuing from the nonhermiticity induced by the spacetime curvature. For conformally flat coordinates in Eq. (3), the Hamiltonian in Eq. (5) can also be recast as

$$
i\gamma^0 \partial_0 \left(\sqrt{\Omega}\,\psi\right) + i\gamma^1 \partial_1 \left(\sqrt{\Omega}\,\psi\right) = M\Omega \left(\sqrt{\Omega}\,\psi\right),
\tag{24}
$$

where the nonunitary evolution becomes explicit. This equation describes the unitary evolution of the field $\widetilde{\psi} = \sqrt{\Omega}\,\psi$ in flat spacetime, given by $i\partial_0 \widetilde{\psi} = -i\gamma_0\gamma^1 \partial_1 \widetilde{\psi} + \widetilde{M}\gamma_0 \widetilde{\psi}$, where the renormalized mass $\widetilde{M} = M\Omega$ depends on the spacetime coordinates. Regularizing on a discrete lattice $x = na$ and discrete time $t = mb$ yields

$$
\frac{i}{2b}\left(\sqrt{\Omega_{n,m+1}}\psi_{n,m+1} - \sqrt{\Omega_{n,m-1}}\psi_{n,m-1}\right)\gamma^0 + \frac{i}{2a}\left(\sqrt{\Omega_{n+1,m}}\psi_{n+1,m} - \sqrt{\Omega_{n-1,m}}\psi_{n-1,m}\right)\gamma^1
$$
$$
= M\Omega_{n,m}^{3/2}\psi_{n,m},
\tag{25}
$$

with $\Omega_{n,m} = \Omega(na, mb)$, and $\psi_{n,m} = \psi(na, mb)$.

In the massless case, the field in flat spacetime evolves as plane waves as $\widetilde{\psi} = e^{i(\omega t + kx)}\chi$ with momentum $k$, energy $\omega = \pm k$, and with $\chi$ the eigenvector of $\gamma_0\gamma^1$ with eigenvalues $\pm 1$. Hence, a massless Dirac field in curved spacetime evolves like

$$
\psi = \frac{e^{i(\omega t + kx)}}{\sqrt{\Omega}}\,\chi,
\tag{26}
$$

which is nonunitary for $\partial_0 \Omega \neq 0$. In the massive case $M \neq 0$, the renormalized mass depends explicitly on the spacetime coordinates, and thus the evolution of the Dirac field is not universal but depends on the details of the metric.

The $\widetilde{\psi}$ describes a Dirac field in flat spacetime and, as such, exhibits unitary evolution and a hermitian Hamiltonian. Conversely, the field $\psi$ describes a Dirac field in curved spacetime and exhibits a time evolution that is not necessarily unitary with a Hamiltonian that is not necessarily hermitian or pseudohermitian. The relation between the two fields is determined by the metric tensor. Figure 4 shows the duality between the field $\widetilde{\psi}$ with mass $\widetilde{M}$ and the field $\psi$ with mass $M$, in the simplest case where the fields are massless $\widetilde{M} = M = 0$. Note that the duality between the Dirac field in flat spacetime and curved spacetime expressed by Eq. (24) is a general property of conformally flat coordinates in Eq. (3) that holds in the massless and massive cases.

For diagonal coordinates in Eq. (13), the Hamiltonian in Eq. (15) can also be recast as

$$
i\sqrt{\beta}\,\partial_0 \left(\sqrt{\beta}\,\psi\right) + i\gamma_0\gamma^1\sqrt{\alpha}\,\partial_1 \left(\sqrt{\alpha}\,\psi\right) = M\gamma_0\alpha\beta\psi,
\tag{27}
$$

which generalizes Eq. (24).

Notably, Eq. (26) expresses the relation between spacetime curvature and two main phenomena in nonhermitian physics: skin modes and nonunitary evolution (gain and loss) [6,7]. To understand the relation between curvature and nonhermitian skin modes, consider a conformal metric which is monotone in space $\partial_1 \Omega > 0$ (or $\partial_1 \Omega < 0$). In the massless case, the Dirac field in Eq. (26) gives $|\psi|^2 = 1/\Omega$: consequently, the probability density is also monotone in space $\partial_1 |\psi|^2 < 0$ (or $\partial_1 |\psi|^2 > 0$). On a finite patch of spacetime, this mandates that the eigenmodes of the lattice Hamiltonian localize at one of the boundaries of the finite lattice: This is the celebrated nonhermitian skin effect in nonhermitian physics, which appears prominently in the Hatano-Nelson model [39, 40, 44] and in other lattice Hamiltonians with non-reciprocal hopping terms. On the other hand, to understand the relation between curvature and nonunitary evolution, consider a conformal metric which is monotone in time $\partial_0 \Omega > 0$

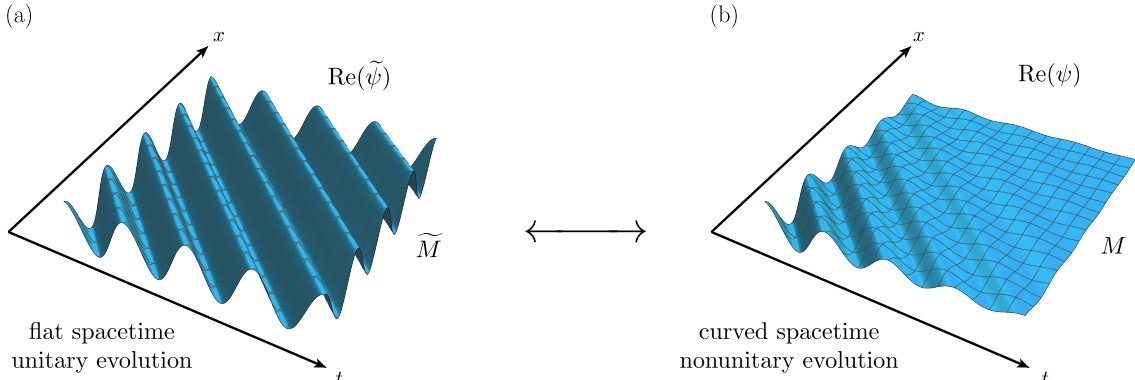

Figure 4: The duality between the unitary evolution of the field $\widetilde{\psi}$ with mass $\widetilde{M}$ in flat spacetime and the nonunitary evolution of the field $\psi$ with mass $M$ in curved spacetime. The relation between the two fields is determined by the metric tensor by $\widetilde{\psi} = \sqrt{\Omega}\,\psi$ and $\widetilde{M} = M\alpha$. For simplicity, the plots show only the real part of the field in the case $\widetilde{M} = M = 0$, with a curved spacetime corresponding to the Weyl metric with $\Omega = e^{rt+qx}$, giving $\widetilde{\psi} = e^{rt+qx}\,\psi$ with $\psi \propto e^{i(\omega t + kx)}$ (plane waves). Note that the field $\psi$ (b) shows both nonunitary time evolution (with the probability density decaying exponentially in time) and the nonhermitian skin effect (with the probability density decaying exponentially in space). In the massive case $M \neq 0$, the renormalized mass $\widetilde{M}$ depends explicitly on the spacetime coordinates.

(or $\partial_0\Omega < 0$). In the massless case, the Dirac field $|\psi|^2 = 1/\Omega$ mandates that the probability density is also monotone in time $\partial_0|\psi|^2 < 0$ (or $\partial_0|\psi|^2 > 0$), i.e., it exhibits nonunitary time evolution. Hence, the nonhermitian skin effect corresponds to a nonzero space dependence of the metric. Conversely, the nonunitary evolution corresponds to a nonzero time dependence of the metric. Hence, the two effects are dual to each other with respect to the exchange of the space and time coordinates.

## 6 Discussion

The relation between spacetime curvature and nonhermitian effect is exemplified by considering the Weyl metric in conformally flat coordinates [see Table 3], which describes a conformal scale expansion or contraction, i.e., a transformation changing the proper distances at each point by the factor $e^{rt+qx}$. In this case Eq. (26) gives $|\psi|^2 = e^{-(rt+qx)}$. This corresponds to a nonhermitian loss for time expansions $r > 0$ and a nonhermitian gain for time contractions $r < 0$. Hence, the nonhermitian loss and gain correspond to the expansion or contraction of the spacetime background. The same considerations can be made for any conformally flat coordinates as in Eq. (3), where the time evolution takes the form of Eq. (24). The Weyl metric also allows one to understand the nonhermitian skin effect in terms of spacetime curvature. For $q > 0$, the probability density of the Dirac field increases towards the left boundary, while for $q < 0$ it increases towards the right boundary of a finite spacetime patch, such that the eigenmodes become localized at one of the boundaries of the discrete lattice under open boundary conditions. Again, the same considerations apply for any conformally flat coordinates as in Eq. (3).

Both the Rindler and de Sitter metrics in diagonal (but not conformally flat) coordinates defined in Table 6 exhibit a coordinate singularity, respectively at $x = 0$ and $x = 1/q$. These singularities sit respectively at the lattice sites $n = 0$ and $n = 1/(aq)$, assuming $1/(aq) \in \mathbb{N}$

in the second case. A defining property of these coordinate singularities is that, when a particle reaches the singularity, there is no way for the particle to escape, no matter the energy of the particle. In the language of the lattice, this point-of-no-return property translates to the fact that the "hopping" amplitude defined in Eq. (20) vanishes between the lattice sites $n = 0$ and $n = 1$ for the Rindler metric, and between the lattice sites $n = 1/(aq) - 1$ and $n = 1/(aq)$ for the de Sitter metric. Hence, the lattice points corresponding to the singularity become completely uncoupled from the rest of the lattice. This corresponds to the presence of a localized zero-energy mode for both the massless and the massive cases. The localized zero-energy mode is visible in Fig. 1(a) and Fig. 1(b) at $x = 0$ for the Rindler metric, and in Fig. 1(c) and Fig. 1(d) at $x = N$ for the de Sitter metric (we choose the parameter $q$ in order to match the length of the lattice $q = 1/N$). The anti-de Sitter metric in Table 6 does not exhibit a singularity. The energy spectrum is gapless in the massless case, and gapped in the massive case, as visible in Fig. 1(d) and Fig. 1(e). The Rindler metric and generalized time-dependent Rindler-like metrics in diagonal coordinates (but not conformally flat) in Table 6 yield hermitian Hamiltonians with unitary time evolutions. Similarly, the de Sitter and anti-de Sitter metrics yield pseudohermitian Hamiltonians with unbroken $\mathcal{PT}$-symmetry, again with unitary time evolutions.

There are several physical systems that simulate the Dirac equation in curved spacetime regularized on a lattice. In essence, the regularized Hamiltonians are discrete tight-binding Hamiltonians with space-dependent and/or time-dependent hopping amplitudes and on-site energies, in combination with nonhermitian effects such as dissipation [6, 7]. Tight-binding Hamiltonians with controllable hoppings and on-site energies are effectively simulated by arrays or lattices of atoms deposited on a surface [45, 46], arrays of quantum dots [47], cold atoms in optical lattices [18, 48–50], photonic crystals [51], superconducting quantum circuits [52–54], topologically nontrivial stripes [55], and exciton-polariton condensates in artificial lattices [56]. Controlling the hopping amplitudes corresponds to controlling the distances and overlaps between contiguous localized states on the lattice, while on-site energies correspond to the presence of a potential.

# 7 Conclusion

In conclusion, I established here a direct correspondence between curved spacetime geometry and nonhermitian physics by analyzing the Dirac equation in 1+1D spacetimes and its lattice regularization, unveiling the geometric origin for nonhermitian phenomena such as nonunitary time evolutions and the skin effect. Specifically, I have shown that for time-independent spacetime coordinates, the lattice Hamiltonians are pseudohermitian and exhibit unbroken $\mathcal{PT}$-symmetry with real spectra and unitary time evolution. In contrast, for time-dependent spacetime coordinates, the lattice Hamiltonians break pseudohermicity, leading to nonunitary evolutions. Similarly, I found that for space-dependent spacetime coordinates, the lattice Hamiltonians exhibit asymmetric "nonreciprocal" hoppings on the lattice that localize eigenmodes at one boundary. Hence, nonhermitian gain or loss and nonunitary time-evolutions are induced by temporal gradients in the spacetime curvature, while spatially-asymmetric nonhermitian terms and the skin effect are induced by spatial gradients. In other words, nonhermitian gain and loss phenomena and the skin effect can be considered the lattice counterparts of expansion and contraction of spacetime. This provides a geometric interpretation of the nonhermitian skin effect and of nonunitary time evolution, and unveils an unexpected connection between gravity and nonhermitian physics.

Finally, this work suggests a broad implication on the nature of the physical reality: Nature may not necessarily be hermitian on a local scale, and the whole universe may, in fact, be a

closed nonhermitian system.

## Acknowledgments

This work was partially supported by the Japan Science and Technology Agency (JST) of the Ministry of Education, Culture, Sports, Science and Technology (MEXT), JST CREST Grant. No. JP-MJCR19T2, the Japan Society for the Promotion of Science (JSPS) Grant-in-Aid for Early-Career Scientists Grants No. 23K13028 and No. 20K14375, Grant-in-Aid for Transformative Research Areas (A) KAKENHI Grant No. 22H05111, and Grant-in-Aid for Transformative Research Areas (B) KAKENHI Grant No. 24H00826.

# Appendices

## A  Another regularization approach

Substituting the spatial derivatives with finite differences $\partial_1 \psi \approx \frac{1}{2a}(\psi_{n+1} - \psi_{n-1})$ in Eq. (14) yields

$$
\mathrm{i}\partial_0 \psi_n = -\frac{\mathrm{i}}{2a}\frac{\alpha_n}{\beta_n}\gamma_0\gamma^1(\psi_{n+1} - \psi_{n-1}) - \frac{\mathrm{i}}{2}\frac{\partial_0 \beta_n}{\beta_n}\psi_n - \frac{\mathrm{i}}{4a}\left(\frac{\alpha_{n+1}}{\beta_n} - \frac{\alpha_{n-1}}{\beta_n}\right)\gamma_0\gamma^1\psi_n + M\alpha_n\gamma_0\psi_n,
\tag{28}
$$

and

$$
\mathcal{H} = a\sum_n -\frac{\mathrm{i}}{2a}\left(\frac{\alpha_n}{\beta_n}\psi_n^\dagger\gamma_0\gamma^1\psi_{n+1} - \frac{\alpha_{n+1}}{\beta_{n+1}}\psi_{n+1}^\dagger\gamma_0\gamma^1\psi_n\right)
$$
$$
-\frac{\mathrm{i}}{2}\frac{\partial_0 \beta_n}{\beta_n}\psi_n^\dagger\psi_n - \frac{\mathrm{i}}{4a}\left(\frac{\alpha_{n+1}}{\beta_n} - \frac{\alpha_{n-1}}{\beta_n}\right)\psi_n^\dagger\gamma_0\gamma^1\psi_n + M\alpha_n\psi_n^\dagger\gamma_0\psi_n,
\tag{29}
$$

which is nonhermitian unless $\partial_1 \alpha = \partial_1 \beta = 0$ and $\partial_0 \beta = 0$, in which case one gets

$$
\mathcal{H} = a\sum_n -\frac{\mathrm{i}}{2a}\frac{\alpha_1}{\beta_1}\left(\psi_n^\dagger\gamma_0\gamma^1\psi_{n+1} - \psi_{n+1}^\dagger\gamma_0\gamma^1\psi_n\right) + M\alpha_1\psi_n^\dagger\gamma_0\psi_n.
\tag{30}
$$

Hence, even for simple metrics such as $\mathrm{d}s^2 = \alpha^2\mathrm{d}t^2 - \mathrm{d}x^2$, using Eq. (6) leads to a hermitian lattice Hamiltonian, while the more standard regularization (finite differences of the wavefunction) leads to nonhermitian lattice Hamiltonians (with the exception of the uniform case).

## B  Calculation of the local density of states (LDOS)

To calculate the local density of states (LDOS) of a given lattice Hamiltonian with real energy spectrum as a function of energy, space, and time, one can first diagonalize the Hamiltonian in position basis and then calculate the LDOS via

$$
\mathrm{LDOS}(x, t, E) = \sum_j \left|\left\langle \psi_j(t)\right|\hat{x}\left|\psi_j(t)\right\rangle\right|^2 \delta(E_j(t) - E),
\tag{31}
$$

where $E_j(t)$ is the $j$-th energy eigenvalue and $\left|\psi_j(t)\right\rangle$ the corresponding eigenstate at time $t$, $\hat{x}$ the position operator, and where the delta function is approximated by

$$
\delta(x) = \frac{1}{\pi}\mathrm{Im}\left(\frac{1}{x - \mathrm{i}\Gamma}\right),
\tag{32}
$$

by taking a conveniently small $\Gamma \to 0^+$. For time-dependent Hamiltonians, the LDOS describes the adiabatic evolution of the spectra in time. These equations apply to Hamiltonians with real energy eigenvalues, i.e., hermitian or pseudohermitian Hamiltonians with unbroken $\mathcal{PT}$-symmetry.

For nonhermitian Hamiltonians with complex energy spectrum, one can consider the LDOS as a function of the real and imaginary parts of the energy [57] by taking

$$\text{LDOS}(x,t,E) = \sum_j \left|\left\langle \psi_j(t) \middle| \hat{x} \middle| \psi_j(t) \right\rangle\right|^2 \delta(\text{Re}(E_j(t) - E))\delta(\text{Im}(E_j(t) - E)), \qquad (33)$$

where $E_j(t)$ is the $j$-th complex energy right eigenvalue and $\left|\psi_j(t)\right\rangle$ the corresponding right eigenstate at time $t$. On the real line, integrating on the imaginary line, one gets

$$\text{rLDOS}(x,t,E) = \sum_j \left|\left\langle \psi_j(t) \middle| \hat{x} \middle| \psi_j(t) \right\rangle\right|^2 \delta(\text{Re}(E_j(t) - E)), \qquad (34)$$

which recovers the usual expression valid for hermitian Hamiltonians. Conversely, on the imaginary line, integrating on the real line yields

$$\text{iLDOS}(x,t,E) = \text{Im}\sum_j \left|\left\langle \psi_j(t) \middle| \hat{x} \middle| \psi_j(t) \right\rangle\right|^2 \delta(\text{Im}(E_j(t) - E)). \qquad (35)$$

These equations apply to Hamiltonians with complex energy eigenvalues, i.e., to pseudohermitian Hamiltonians with no $\mathcal{PT}$-symmetry, or Hamiltonians that are neither hermitian nor pseudohermitian.

In all cases considered, the calculations are performed on a finite lattice of 500 sites with open boundary conditions and lattice parameter $a = 1$, taking $q = 1/N$, $M = 0$ in the massless and $M = 0.5$ in the massive case, respectively. The LDOS is plotted in arbitrary units, normalized to its maximum value (separately on the real and imaginary line), and using the "cubehelix" color scheme [58].

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
