# Peer review of "Metric-induced nonhermitian physics"

_SciPost Physics_

## Round 2 · Author Response

I thank the Referee for their positive evaluation of our work and for their insightful suggestions. Below is a point-by-point answer to the Referee's comments.
1) Following the referee's suggestion, I reorganized the paper, starting with the analysis of the conformally flat coordinates. Moreover, since the lattice regularization breaks the usual invariance upon a change of coordinates, I also included an extended analysis of diagonal coordinates. All other cases (Rindler, de Sitter, anti-de Sitter, Weyl metric) are now derived as special cases in Tables 3 and 6.
2) The Referee rightfully suggests that the conditions under which Hermiticity can hold should be clarified. The conditions for hermicity, pseudohermicity, and other symmetry properties are now derived explicitly for conformally flat coordinates and diagonal coordinates in Section 3, and summarized in Tables 1, 2, 4, and 5. In short, for conformally flat coordinates, pseudohermicity is present only for static (time-independent) coordinates. For diagonal coordinates, pseudohermicity requires only that the spatial part of the metric does not depend on time. Note that pseudohermicity is a property that depends on the coordinates chosen, and not on the spacetime. Obviously, all physical observables are invariant upon a change of coordinates in the usual continuum models. However, when regularizing the spatial dimension on the lattice, this invariance is broken. This point is also discussed in Section 4.
3) What do we learn from the computation of the density of states? In short, the density of states is a measurable quantity in condensed matter systems that allows one to probe nonhermitian effects (such as the skin effect and the presence of singularities). I included a discussion of the relevance of the density of states calculations at the end of Section 3.
In summary, I revised the paper to deal with the conformally flat case and consider other cases as special cases, and I also explicitly derived the conditions for hermicity and pseudohermicity, as suggested by the referee.

---

## Round 2 · List of Changes

The major changes are in Sections 3 and 4.
These two sections address conformally flat and diagonal coordinates separately.
Section 5 has been revised to include a discussion of the nonhermitian skin effect.
Introduction, discussion, and conclusions have been revised accordingly.

---

## Editorial Decision

unknown